# The Melody of Speech: What the Melodic Perception of Speech Reveals about Language Performance and Musical Abilities

Markus Christiner [1,*], Christine Gross [2], Annemarie Seither-Preisler [1] and Peter Schneider [3]

1 Centre for Systematic Musicology, University of Graz, A-8010 Graz, Austria; annemarie.seither-preisler@uni-graz.at
2 Latvian Academy of Music, Jazeps Vitols Latvian Academy of Music, LV-1050 Riga, Latvia; christine.michaela.gross@jvlma.lv
3 Section of Biomagnetism, Department of Neurology, University Hospital Heidelberg, 69120 Heidelberg, Germany; schneider@musicandbrain.de
* Correspondence: markus.christiner@uni-graz.at

**Abstract:** Research has shown that melody not only plays a crucial role in music but also in language acquisition processes. Evidence has been provided that melody helps in retrieving, remembering, and memorizing new language material, while relatively little is known about whether individuals who perceive speech as more melodic than others also benefit in the acquisition of oral languages. In this investigation, we wanted to show which impact the subjective melodic perception of speech has on the pronunciation of unfamiliar foreign languages. We tested 86 participants for how melodic they perceived five unfamiliar languages, for their ability to repeat and pronounce the respective five languages, for their musical abilities, and for their short-term memory (STM). The results revealed that 59 percent of the variance in the language pronunciation tasks could be explained by five predictors: the number of foreign languages spoken, short-term memory capacity, tonal aptitude, melodic singing ability, and how melodic the languages appeared to the participants. Group comparisons showed that individuals who perceived languages as more melodic performed significantly better in all language tasks than those who did not. However, even though we expected musical measures to be related to the melodic perception of foreign languages, we could only detect some correlations to rhythmical and tonal musical aptitude. Overall, the findings of this investigation add a new dimension to language research, which shows that individuals who perceive natural languages to be more melodic than others also retrieve and pronounce utterances more accurately.

**Keywords:** melodic language perception; melodic perception; melody; phonetic; musical abilities; music perception; singing ability



## 1. Introduction

Interdisciplinary research on music and language has become rather diverse over the past two decades. The reason for this development is evident as music and language share a set of characteristics (Jackendoff and Lerdahl 2006). Music and language are based on hierarchical structural aspects, such as the ordering of distinct elements (Jackendoff and Lerdahl 2006; Honing 2011) and consist of tonal and rhythmical features. The similarities between language and music are rather salient on the acoustic level. This becomes particularly obvious if one looks at speech directed to infants. It is rather slow, shows more pitch variation, and is often perceived to be more melodic in its characteristics than adult speech (Kuhl et al. 1997; McMullen and Saffran 2004). Indeed, song and melody are based on discrete pitches, which are sustained over longer durations compared to speech (Deutsch et al. 2011). Even though language and music show many similarities, they are based on different sound systems. Whereas that for music is based on pitches and timbres, the linguistic sound system consists of pitch contrasts, vowels, and consonants (Patel 2007).

In general, various scientific branches that attempt to analyse rhythmic and tonal aspects of music and their relationship to language prosody have emerged (Krumhansl and Keil 1982; Patel 2007; Patel and Daniele 2003). For instance, the pitch structure of music and language have been extensively studied by Jackendoff and Lerdahl (2006). On a syntactic level, language has also been compared to discrete structural elements of music (Honing 2011; Patel 2003). More recently, diverse scientific branches have started looking at potential positive transfer effects from music to language, and vice versa. For the past two decades the scientific community has shown considerable interest in understanding the underlying mechanisms of musical aptitude and musical training. Whereas the latter is associated with achievement and mastery, musical aptitude is compared to potentials that can be seen as a kind of readiness to learn (Gordon 1989; Law and Zentner 2012). It is generally accepted that musical proficiency is comprised of the interactions between acquired and innate musical capacities (Sloboda 2008). More recently, studies on the relationship between music and language have also discussed potential pre-existing abilities, which may be responsible for the link between both faculties (Swaminathan and Schellenberg 2020; Kragness et al. 2021). This addresses transfer effects between music and language, which are not induced by formal musical training.

According to recently published studies, positive relationships between music and language learning have been found on multiple occasions. For instance, music-based training has been suggested to facilitate duration perception in speech (Chobert et al. 2014) and the ability to segment speech (François et al. 2013). Trained musicians seem to detect incongruities in unfamiliar speech better than non-musicians do (Christiner 2020) and musical aptitude has generally been linked to language functions in children and adults (Christiner and Reiterer 2018, 2019; Christiner et al. 2018; Turker et al. 2017; Turker 2019).

Working memory (WM) capacity has been described as a system that enables the storing, manipulating, and maintaining of temporary information (Baddeley 2003). Complex WM capacity has an influence on multiple cognitive domains such as intellectual (Conway et al. 2002, 2003; Engle et al. 1999) and mathematical ability (Schmader and Johns 2003). Therefore, WM capacity has received considerable attention in music and language research and is associated with individual differences in the mastery of first and foreign languages (Baddeley et al. 1998; Dörnyei and Ryan 2015; Majerus et al. 2006; Wen and Skehan 2011). In language research, the subsystem of the WM, the phonological short-term memory (STM), is the most important capacity for observing individual differences in language abilities (Wen and Skehan 2011). STM capacity is related to the ability to remember larger phonological structures and is the most important cognitive capacity that predicts refined language abilities of multilinguals and polyglots (Baddeley et al. 1998). Therefore, if language abilities are assessed, STM capacity should be investigated as well. Whereas, in language research, STM has intensively been studied, in music research, it is different. There is a controversy over whether a "tonal loop" in music as an equivalent of the phonological loop for language capacity exists or not. Although early research suggested a separate storage for tonal and speech material (Salame and Baddeley 1989), more recently it has been shown that the processing of musical and verbal sounds show overlaps (Williamson et al. 2010). Brain research reported that verbal and tonal storage rely on largely overlapping neuronal networks (Koelsch et al. 2009). This may be one reason why STM capacity is associated with enhanced language and with improved musical capacities.

In the past two decades, extensive research on the relationship between music and language has been published in the fields of education and aptitude. These publications mainly aimed at illustrating the positive effects of music on language ability and language learning progress. Several studies have reported a link between musical ability and foreign speech production, such as the ability to pronounce foreign languages (Milovanov et al. 2009; Milovanov and Tervaniemi 2011; Pastuszek-Lipinska 2008). In aptitude research, both tonal and rhythmic musical abilities predicted phonetic skills in the learning of unfamiliar languages. Whereas a tonal subtest, as measured by the AMMA test (Gordon 1989), was



more predictive for adults in the ability to pronounce multiple languages (Christiner 2020), the opposite was found for children, where rhythmic predictors were found to explain enhanced language skills (Swaminathan et al. 2017). Language typology also seems to influence the relationship between language and music. Tone language imitation ability was predicted by tonal aptitude, whereas non-tone language imitation was predicted by rhythmic aptitude (Christiner et al. 2018). Singing, for instance, was found to facilitate the learning of new vocabulary (Ludke et al. 2014) and was often employed as a learning tool in the foreign language classroom for beginners. For example, foreign words were presented and learnt together with a melody (Anton 1990). Singing new words in foreign languages is also assumed to facilitate retaining new utterances more easily (Ludke et al. 2014). The key role for this has often been ascribed to melody (Purnell-Webb and Speelman 2008). Indeed, infants also acquire new utterances much faster when they are sung (Thiessen and Saffran 2009). Melody is also said to serve as a mnemonic with which utterances are stored in the long-term memory (Gordon et al. 2010) and "[ . . . ] seems to act as a path or a cue to evoke [ . . . ]" information (Fonseca-Mora 2000, p. 150). On these grounds, melody not only plays a key role in music but also in language acquisition processes.

Interestingly, while the relationship between language and music has been addressed in various domains, looking at how melodic languages are perceived has largely been neglected. There is some research that focuses on a phenomenon in which spoken utterances are transformed to sound like song, which is achieved by repetition (Deutsch et al. 2011). In a series of experiments, the researchers concluded that this phenomenon is valid as long as the samples, which were repeatedly provided, were exactly the same. This so-called speech-to-song illusion was also investigated by Margulis et al. (2015), who additionally related their findings to foreign language pronunciation skills. They suggested that the speech-to-song illusion occurred more readily when the speech material was more difficult to pronounce (Margulis et al. 2015). However, in this investigation, we approached from another direction. As melody helps individuals to remember language material more easily, we wanted to know whether the subjective melodic perception of unfamiliar languages influences individuals' ability to pronounce foreign languages. In this research, the melodic perception of speech describes the listeners subjective impression of how melodic and musical spoken languages appear to be.

We suggest that if melody indeed has such an enormous impact on language functions, the subjective perception of how melodic different languages sound should also have a profound influence on language capacity. This is our first research question (Q1). The second research question (Q2) focuses on the possibility that the melodic perception of speech may be a new predictor, which could partly explain the variances in language performances besides previously found indicators. In this respect, it is crucial to clarify how musical and language capacity can be measured. In addition, as musical abilities contribute to language functions, we also wanted to test whether there is a relationship between measures of musicality and the subjective melodic perception of languages. The latter represents our last research question (Q3).

### 1.1. Assessing Musical Abilities

For measuring musical abilities, various approved musicality tests are available. Most of them are perception tasks, which at least consist of rhythmic and tonal subtests. The Advanced Measures of Music Audiation (AMMA) test developed by Gordon (1989) has been used in multiple investigations and reliably measures the ability to discriminate tonal and rhythmic changes in paired musical statements. In addition, interdisciplinary research that used the AMMA test and compared tonal and rhythmic abilities to phonetic language abilities is available (Christiner and Reiterer 2013, 2015, 2019; Turker et al. 2017). However, increasingly more studies show contradictory results when the relationship between music perception (pitch discrimination) and production is investigated (Berkowska and Bella 2009). While some studies have reported a relationship between the production and perception of music (Demorest et al. 2015; Demorest and Pfordresher 2015), others have not (Loui

et al. 2009; Pfordresher and Mantell 2014; Tremblay-Champoux et al. 2010). Therefore, if musical abilities are assessed, the inclusion of music performance and music perception measures will more reliably illustrate the musical capacities of individuals. Measuring musical performances is achieved best by introducing singing tasks. This has the advantage that non-musicians who do not play a musical instrument can participate in the research as well (Dalla Bella et al. 2007).

In general, singing tasks are subdivided into two main categories: imitation (repeating new, unfamiliar melodies or songs) and tasks where participants have to sing familiar songs. While imitation tasks are often used for advanced singers, familiar song singing tasks are often targeted at non-musicians (Dalla Bella et al. 2007, 2009). The assessment of singing performance can be carried out by means of computerized methods, which focus on pitch accuracy (Salvador 2010). Another option is to use rating scales where the performances are evaluated based on specific criteria by experts (Hornbach and Taggart 2005; Rutkowski and Miller 2002). Rating scales can be used in a rather flexible way and adapted to evaluate specific rating criteria (Larrouy-Maestri et al. 2013), and longer sequences can easily be assessed (Christiner 2020). The latter approach has been chosen in this study.

### 1.2. Assessing Pronunciation Skills and the Melodic Perception of Speech

Measuring individual differences in the ability to pronounce new words can be achieved best by using unfamiliar short sequences of language stimuli that individuals are instructed to repeat. Subsequently, their performances will be assessed by experts or native speakers. These measurements are of high ecological validity because they simulate a foreign language situation in which new words or phrases are learned. In addition, the same language stimuli can easily be rated for how melodic they appear to listeners.

Using unfamiliar utterances as test stimuli, however, has more advantages. One is that individual differences in the performances also vary depending on foreign language capacity. Therefore, using language stimuli that are unfamiliar to individuals ensures that educational influences on performances are reduced—a common approach, which has successfully been used in previous investigations (Christiner and Reiterer 2013; Christiner and Reiterer 2015; Christiner and Reiterer 2018). Another benefit is that sociolinguistic influences are minimized and reduced. This means that neither the impact of the message of the content, nor the recognition of particular speech styles and social identities, can trigger certain likes, dislikes, or social categorizations and infer that speakers possess particular personality attributes (Giles and Billings 2014). Since recently the nature of short sequences of unfamiliar languages as test stimuli have been investigated in more detail, this represents another advantage. For instance, factor analysis revealed that typologically different short sequences of language stimuli load onto the same factor, which suggests that short sequences of unfamiliar speech measure general pronunciation ability, even if they are typologically different (Christiner 2020). This finding has two crucial implications. One is that imitation tasks of different languages represent a general aptitude and pronunciation measurement. The second is that many languages can be used to create a single measurement, which represents a more reliable concept to measure pronunciation skills.

Regarding approaches towards measuring the melodic perception of speech from a musicological point of view, there are further good reasons to use unfamiliar utterances. One is that, in initial foreign language learning situations, language input is rather meaningless and may force naïve listeners to treat language stimuli similar to musical statements (Milovanov et al. 2009). This suggests that more music-resembling language features (e.g., speech melody) are in the foreground of the speech material to which individuals are exposed to. Indeed, natural pitch modulations in spoken language have a lot in common with tone transitions in musical melodies (Oechslin et al. 2010), and brain research provided evidence that prosodic information is predominantly processed in the right area of the auditory cortex (Meyer et al. 2002) when linguistic information is rather poor in content (Perkins et al. 1996). In consideration of the criteria and measurements, which were discussed in the former two sections, the research design was created.

Since we aimed at providing information about whether individuals who perceive languages to be more melodical than others also perform better in the pronunciation of unfamiliar languages (Q1), we used measurements based on a previous test design. We selected four samples in five different languages. Subjects were tested for how well they could retrieve the samples as well as how melodic the samples appeared to them. As it is plausible that the language material provides information about general phonetic and pronunciation ability (Christiner 2020), we analyzed the five languages separately and as a single measurement. Since we also wanted to investigate whether there is a relationship between musical measures and how melodic languages are subjectively perceived (Q2), we decided to include different tests of musical abilities: the AMMA test as a music perception task and singing as a music performance task. In addition, we hired professionals, amateurs, and non-musicians for this investigation to create further musical categories of different training status. We assumed that if melody has an impact on language capacities, individuals who perceive languages to be more melodic will also perform better in the language performance tasks and probably also in the music measurements. Finally, we also wanted to know whether the characteristic of how melodic languages appear to individuals is also a predictor for explaining the variance in the language performance beside previously found indicators, such as STM capacity, singing ability, the number of foreign languages, and musical aptitude (Q3).

## 2. Materials and Methods

### 2.1. Participants

For this investigation we recruited 86 participants. All of them voluntarily participated in the study, and informed consent was obtained from all subjects involved in the study. None of them reported to have any hearing or other impairments. In this study, 36 participants were male, and 50 participants were female. The mean age was $M = 34.53$, $SD = 11.51$.

### 2.2. Educational Status

The participants' educational status was entered according to the educational status that had been completed at the testing time. The results revealed that 42 participants completed secondary academic school (general qualification for university entrance), 15 had a bachelor's degree, 26 had a master's or a doctoral degree, and 3 did not indicate their educational status.

### 2.3. Musical Measurements

2.3.1. Musical Background

The participants reported their musical activity, the musical instruments played, and had to label themselves to be either non-musicians, amateurs, or professional musicians. It was explained that being a non-musician meant that they are not capable of playing a musical instrument. In addition, they were also asked whether they no longer train or play musical instruments despite having trained for years. The latter were not included in this study. Being an amateur meant that they should be capable of playing one or more musical instruments, as well as that they play musical instruments occasionally, but not professionally. Being a professional musician included that the participants played regularly publicly as members of an orchestra at least for two years, or studied music for three semesters, or were music teachers. The results showed that, based on the definitions, 30 were classified as professional musicians, 21 as amateurs, and 35 as non-musicians. We also collected information about the number of instruments the amateurs and musicians played. The responses showed that 22 played one, 18 played two, 2 played three, 4 played four, 5 played five, and 1 played seven instruments.

2.3.2. Musical Aptitude: Advanced Measures of Music Audiation

The AMMA test measures the participants' potential to discriminate paired musical statements that are either different or the same. Participants have to choose between three different conditions such as whether the paired musical statements are the same or include rhythmical or tonal change. The paired musical statements are embedded in one test design where either tonal, rhythmic, or no changes can occur. This test is usually targeted at university music and non-music majors and high school students and is an aptitude test. The test consists of 33 items. The first three are familiarization tasks and were excluded from the final analysis.

2.3.3. Singing Ability

Singing ability was tested and measured in two different ways. One task was to sing the familiar song "Happy Birthday," since this is usually targeted at both professionals and non-professionals (Dalla Bella et al. 2007; Dalla Bella and Berkowskaa 2009; Christiner 2020; Christiner and Reiterer 2013, 2019; Christiner et al. 2018).

The second singing task was more complex. It consisted of two imitation tasks where parts of an unfamiliar song had to be learnt in a rather short period of time. Therefore, we used an adapted version of a singing task, which we had successfully used in previous research (Christiner 2020; Christiner and Reiterer 2013). The adaptation meant omitting the longest sequence. Based on previous findings we knew that participants managed to sing the short sequences of the two parts of the song no matter whether they were musicians or not (Christiner 2020). The aim of this task was to actively engage the participants in a singing learning condition to measure their singing ability. This learning condition was split into two parts, which became increasingly difficult. The participants had to sing the original part of the song after they had listened to the original sound file three times (lyrics were provided). Singing with lyrics demonstrates the full vocal repertoire and makes it possible to address more rating criteria (Larrouy-Maestri et al. 2013). The lyrics and the notes of the short sequence of the song are provided in the supplement (Figure S1). The original part of the song was accompanied by musical instruments. However, the participants had to sing the song for the recording without background music and only from memory as well as possible. The participants were further instructed to repeat the song in a key which they found comfortable, as key did not play a role in the final ratings.

The singing performances of the participants were rated and evaluated by singing experts (two male and two female raters) who received some compensation for their work. The procedure had successfully been used and tested in previous studies (Christiner and Reiterer 2013). The rating criteria for both songs were melodic and rhythmic ability.

Therefore, the raters were instructed to evaluate how well the participants were able to repeat the new melodies of the two imitation tasks and how well they sang the melody of the song "Happy Birthday." For the rhythmic ratings, they were asked to evaluate how well the participants were able to maintain the original rhythms of the two imitation tasks and how accurate the rhythmic structure of "Happy Birthday" appeared to the experts. Therefore, the raters received a login and performed the ratings online. They had to evaluate all performances of all participants. Since it was not possible to do the ratings within a single sitting, the ratings consisted of two main sections and three subsections. The main sections were divided into the rhythmic and melodic ratings, and the subsections were comprised of the two imitation tasks and "Happy Birthday." We did not mix rhythmic and melodic ratings since we wanted the raters to focus on only one element during the rating process before they went on to the next rating criterion. The first six performances in all rating sections were familiarisation tasks. Therefore, we took samples of participants who had scored high, average, and low in previous investigations. The performances of the participants in this investigation were presented in randomized order. The rating scales ranged from 0, "min," to 10, "max." Based on the ratings, two scores, one for melodic performance (melodic singing ability) and one for rhythmic performance (rhythmic singing ability), were determined. Both scores were compound measures of the ratings for the two

singing tasks, respectively. This approach was based on the findings of former research where we had assessed the nature of the same singing ratings we used in this investigation. There, factor analysis showed that familiar and unfamiliar song singing tasks belong to one factor. This was also shown to be consistent after a follow-up reliability analysis (Christiner 2020). Therefore, we also applied an interrater reliability by means of using Cronbach's alpha coefficients to assess the internal consistency of the performances of our raters. This was determined for melodic singing ability and rhythmic singing ability. For interrater reliability, Cronbach's alpha coefficients were determined as well for melodic singing ability as for rhythmic singing ability. For melody, the Cronbach's alpha coefficient was 0.95, and for rhythm it was 0.93. Thus, interrater reliability was very high in both cases.

*2.4. Language Measurements*

2.4.1. Language Background

The participants were all German native speakers who had grown up and who had acquired foreign languages in the educational settings. Given participants' experience with foreign languages, we addressed the number of foreign languages (no. of FL) spoken by participants in the study. The results showed that 10 participants did not speak a second language, even though they were taught English in school; 28 participants reported to speak one foreign language; 36 indicated to speak two foreign languages; and 12 participants reported to speak three foreign languages. The foreign languages participants reported to be speaking were English, French, Spanish, Hungarian, Italian, Swedish, Afrikaans, and Dutch. None of the participants spoke one of these languages, which were included in the research design: Chinese, Japanese, Tagalog, Thai, and Russian.

2.4.2. Language Performance

For the language performance tasks, stimuli consisting of eleven syllables that were spoken by two male and two female native speakers were created for each of the selected languages (Chinese, Japanese, Tagalog, Thai, and Russian). The reason for choosing these languages was that they are rarely spoken in Germany. The selection of male and female voices in each language aimed to minimize gender's effects on the language performances and the melodic language ratings of the participants. In addition, prosodic and paralinguistic features of the voice, such as speech rate and voice quality, were kept constant across languages. Therefore, we used only declarative statements and no questions to avoid extreme differences in intonation. In this respect, we also paid particular attention to keep the speech rate as similar as possible across languages.

The participants had to repeat four simple phrases or sentences (statements) of eleven syllables for each of the five languages in a single sitting. This resembles foreign language learning and reveals how well and fast individuals acquire unfamiliar languages. First, the procedure included a familiarization task where the participants were instructed to imitate language samples, which were unfamiliar to them as well. For this task, four samples in the Slovak language were used. Each sample was played three times separated by a pause of 50 ms. Next, the participants had to repeat the sentence or phrase and try to imitate the accent as best they could. In the testing condition, each of the imitations in the respective languages was recorded and the next language sample was played. The participants wore headphones with an integrated microphone (Beyerdynamic, DT 290) while their language performance was recorded. Every single recording of the participants' performances was cut, checked, its loudness normalized, and subsequently uploaded to an online rating platform, where the performance for each language was evaluated by native speakers of the respective languages. The raters were selected based on two criteria. One was that they should be native speakers of the respective five languages. The other was that they should have a professional linguistic background. Therefore, we hired the raters at different universities and offered them some compensation for their work. The rating scale ranged from 0, "min," to 10, "max," and the raters were instructed to evaluate the overall performance of each of the language stimulus in the respective languages. We

consulted four raters for Russian, six for Japanese, five for Chinese, and five for Thai. A reliability analysis was performed for each of the language ratings. This procedure had been used several times in previous research to assess the internal relationship of the ratings (Christiner and Reiterer 2013; Christiner et al. 2018; Christiner 2020). Cronbach's alpha coefficients were calculated recommending a minimum score of 0.7 in general (Field 2009). The interrater reliability of each language was rather high and varied between 0.86 and 0.93.

### 2.4.3. Melodic Language Ratings

For the melodic language ratings, the participants were familiarized with their tasks. Therefore, they had to listen to samples in Slovak and Farsi. The participants were instructed to indicate how melodic and tuneful the language samples occurred to them. Under the testing conditions, the participants were listening to the four different samples in each of the five languages in a row, and after they had indicated their judgements of how melodic the respective languages appeared to them, they were listening to the four samples of the next language. The samples the participants rated were the same that they had repeated before. The only information the participants received about the languages was that the four samples always represented examples of the same language. The reason why we refused to name the languages was that we wanted to minimize bias regarding sociolinguistic aspects, such as positive or negative associations with and attitudes toward one of the languages we had selected. The rating scale ranged from 0, "min," to 10, "max," where 10 was the highest score (very melodic) and 0 the lowest score (not melodic).

### 2.5. *Short-Term Memory Measurement*

In order to test the STM capacity of the participants, the Wechsler Digit Span (Wechsler 1939) was used. This well-known test is composed of a forward digit span and a backward digit span. The test was programmed online, and the stimulus was presented auditorily. This test consists of a digit span forward and a digit span backward sub-test. The participants had to repeat a steadily increasing sequence of digits in either a forward or a backward order. The sequences ranged from 3 to 9 digits for the forward version and from 2 to 8 digits for the backward version.

### 2.6. *Testing Procedure*

The participants were first instructed to fill in the questionnaires on background information (language, music, and school education) at home before further behavioral testing took place. The idea was to pre-select participants since one criterion was that the languages should be unfamiliar to the participants. This should guarantee that the participants had equal chances to perform well in the language tasks and to make sure that the ratings were not influenced by sociolinguistic aspects. After the participants had provided information about their language and music background, we invited them to our lab. Testing was split up into two different sessions as a single sitting would have been too long. The first session included verifying the background information from the questionnaire, the AMMA musicality test, the singing tasks, and STM measurement. Even though the participants were in the lab, the tasks were performed online as we wanted to make sure that conditions were the same for everyone. The first session lasted around 60 min. In the second session, the participants performed the online language pronunciation tasks in Chinese, Japanese, Tagalog, Thai, and Russian in our lab. However, the participants were assisted with the recordings to make sure that they could focus on their tasks. Finally, the participants listened to the same language stimuli, which they had repeated before, again and had to indicate how melodic the individual languages appeared to them. On average, the second session lasted 70 min.

Although it was necessary to invite the participants to our lab, the native speakers who performed the language ratings, and the singing experts who evaluated the singing performances, rated the performances of the participants online. The rating criteria were precisely described on the online platform respective to their tasks. In addition, each rater was instructed online either in person or in writing by one of our experimenters before they began.

Since we created several variables, we produced a list of abbreviations for better illustration (see Table 1).

**Table 1.** Abbreviations.

| Abbreviation | Meaning |
| --- | --- |
| AMMA | Advanced Measures of Music Audiation |
| AMMA rhythm | Rhythmic AMMA score |
| AMMA tonal | Tonal AMMA score |
| ES | Educational status |
| High melodic LP | High melodic language perceivers |
| Low melodic LP | Low melodic language perceivers |
| Melodic P | Mean of the composite score of all five melodic ratings |
| No of FL | Number of foreign languages spoken |
| P | Perception |
| PR | Pronunciation score |
| PR total | Mean composite score of all five language performance measurements |
| STM | Short-term memory |

### 2.7. Statistical Analysis and Procedure

The statistical analysis is subdivided into different approaches. First, we analyzed whether the language performances were related to the melodic perception of speech, the musical measurements, STM capacity, the number of foreign languages learnt, and the educational status by means of correlational and regression analysis. In addition, we analyzed whether meaningful groups could be created based on the melodic language ratings. Therefore, we applied a cluster analysis to create clusters, which resulted in two distinct groups: high and low melodic language perceivers (see Supplement Table S7). We also ran a series of *t*-tests for independent samples and ANOVAs. The latter aimed at testing the musical capacity of the participants to illustrate whether their own definition as "professionals," "amateurs," and "non-musicians" was also reflected in the musical measurements (see Supplement Tables S8 and S9). The final analysis focuses on whether there are interactions between the musical status of the participants and the high and low melodic language perceivers with regard to language performance. Therefore, we performed a two-way ANOVA. The dependent variable was the "pronunciation total score," which is comprised of the means of all five language scores. The independent variables were musical status (professional, amateur, and non-musician) and melodic language perception (high and low melodic language perceivers).

### 3. Results

### 3.1. Descriptives of the Measurements

In Table 2 below the descriptives of the variables under consideration are provided.

**Table 2.** Descriptives of the variables of this investigation.

| Variables | Mean (*M*) | Standard Deviation (*SD*) |
|---|---|---|
| Melodic ratings for Chinese | 5.85 | 2.39 |
| Melodic ratings for Japanese | 5.83 | 2.45 |
| Melodic ratings for Russian | 5.73 | 2.24 |
| Melodic ratings for Tagalog | 6.91 | 1.92 |
| Melodic ratings for Thai | 4.70 | 2.11 |
| **Melodic perception (Melodic P)** | 5.80 | 1.37 |
| Chinese pronunciation (PR) | 2.37 | 0.83 |
| Japanese pronunciation (PR) | 4.82 | 1.38 |
| Russian pronunciation (PR) | 3.60 | 1.35 |
| Tagalog pronunciation (PR) | 2.36 | 1.18 |
| Thai pronunciation (PR) | 1.64 | 0.73 |
| **Pronunciation (PR) total** | 2.96 | 0.89 |
| AMMA rhythm | 28.70 | 4.26 |
| AMMA tonal | 25.86 | 5.10 |
| Melodic singing ability | 5.98 | 1.50 |
| Rhythmic singing ability | 6.77 | 1.18 |
| Short-term memory (STM) | 15.23 | 3.84 |

*3.2. Statistical Results 1: Relationships among the Selected Variables (Correlations and Regression Models)*

Correlational analyses were applied in order to provide information about the relationship between the variables of interest. We used the composite scores of the melodic language ratings and the language performances for the main analysis as this represents a more reliable concept (Christiner 2020). Results are provided in Table 3 below:

**Table 3.** Simple associations of the variables of this investigation.

| Variable | Melodic P | Melodic Singing Ability | Rhythmic Singing Ability | AMMA Tonal | AMMA Rhythm | STM | ES | No. of FL |
|---|---|---|---|---|---|---|---|---|
| PR total | 0.466 ** | 0.512 ** | 0.501 ** | 0.401 ** | 0.324 ** | 0.503 ** | 0.231 * | 0.503 ** |
| Melodic P | | 0.168 | 0.181 | 0.203 | 0.225 * | 0.235 * | 0.309 ** | 0.304 ** |
| Melodic singing ability | | | 0.964 ** | 0.434 ** | 0.446 ** | 0.244 * | 0.283 ** | 0.370 ** |
| Rhythmic singing ability | | | | 0.419 ** | 0.417 ** | 0.259 * | 0.254 * | 0.370 ** |
| AMMA tonal | | | | | 0.789 ** | 0.120 | 0.127 | 0.208 |
| AMMA rhythm | | | | | | 0.194 | 0.048 | 0.227 * |
| STM | | | | | | | 0.079 | 0.201 |
| ES | | | | | | | | 0.367 ** |

*Note.* PR = pronunciation. P = perception. T = tonal. R = rhythmic. STM = short-term memory. ES = educational status. No. of FL = number of foreign languages. * $p < 0.05$ (uncorrected, two-tailed). ** $p < 0.001$ (uncorrected, two-tailed).

After inspection of the correlation matrix, a multiple linear regression analysis was performed. In the regression models, all variables that correlated to the PR total score (the dependent variable) were entered into a multiple linear regression as independent variables. The independent variables were included in the multiple linear regression models only if a probability of *F*-change < 0.05 was given. Therefore, a stepwise method was chosen, and the ordering of the variables was based on purely mathematical decisions. The results revealed that around 59 percent of the variance in the performances could be explained by six predictors: no. of FL, STM, AMMA tonal, melodic P (how melodic the languages were rated), and melodic singing ability (how well the participants sang melodies).

Table 4 below illustrates the regression models.

**Table 4.** Multiple regression models explaining the variance in pronunciation (PR) total.

| Predictor | Partial Correlation (*pr*) | *p*-Value |
|---|---|---|
| Step 1: $R = 0.52$, $F(1, 80) = 30.25$, $p < 0.001$ | | |
| No. of FL (foreign lang.) | 0.52 | <0.001 |
| Step 2: $R = 0.65$, $F(1, 79) = 19.73$, $p < 0.001$ | | |
| No. of FL (foreign lang.) | 0.49 | <0.001 |
| STM | 0.45 | <0.001 |
| Step 3: $R = 0.71$, $F(1, 78) = 12.41$, $p < 0.001$ | | |
| No. of FL (foreign lang.) | 0.46 | <0.001 |
| STM | 0.44 | <0.001 |
| AMMA tonal | 0.37 | <0.001 |
| Step 4: $R = 0.74$, $F(1, 77) = 8.79$, $p = 0.004$ | | |
| No. of FL (foreign lang.) | 0.41 | <0.001 |
| STM | 0.42 | <0.001 |
| AMMA tonal | 0.34 | 0.002 |
| Melodic P. total | 0.32 | 0.004 |
| Step 5: $R = 0.77$, $F(1, 76) = 6.9$, $p = 0.010$ | | |
| No. of FL (foreign lang.) | 0.33 | |
| STM | 0.40 | <0.001 |
| AMMA tonal | 0.24 | 0.031 |
| Melodic P. total | 0.34 | 0.002 |
| Melodic singing ability | 0.29 | 0.010 |
| Dependent variable: pronunciation (PR) total | | |

*3.3. Statistical Results 2: Group Differences for High vs. Low Melodic Language Perceivers (t-Tests for Independent Samples)*

In order to test whether clusters of groups can be differentiated based on the melodic language ratings of the five languages, a cluster analysis was applied, which resulted in two groups: high melodic LP and low melodic LP (see Supplement Table S7). Since we wanted to clarify which abilities were different in the two melodic perception groups (high melodic LP; low melodic LP), we performed *t*-tests for independent samples for the language-related, music-related, and STM measures. The results revealed that neither the musical measures (melodic singing ability, rhythmic singing ability, AMMA tonal, and AMMA rhythm), nor STM capacity, were statistically different in the two groups (see Table 5). However, the high melodic LP group performed significantly better in all language performance tasks in Chinese, Japanese, Tagalog, Thai, and Russian and consequently also in the PR total, which comprised all five languages. Since there is no previous research available that addresses the melodic perception of languages in this context, for the sake of transparency, we wanted to provide the individual five language scores. The results indicated that the high melodic LP group performed significantly better than the low melodic LP group in all languages scores.

**Table 5.** Independent *t*-tests of the high melodic language perceivers and the low melodic language perceivers.

| Variables | Low Melodic LP: Mean | Low Melodic LP: SE | High Melodic LP: Mean | High Melodic LP: SE | *t* | df | *p* | *r* |
|---|---|---|---|---|---|---|---|---|
| Chinese PR * | 2.11 | 0.12 | 2.62 | 0.12 | −3.02 | 84 | $p < 0.003$ | $r = 0.31$ |
| Japanese PR * | 4.30 | 0.21 | 5.32 | 0.18 | −3.68 | 84 | $p < 0.001$ | $r = 0.37$ |
| Russian PR | 3.20 | 0.19 | 3.97 | 0.21 | −2.75 | 84 | $p < 0.007$ | $r = 0.29$ |
| Tagalog PR * | 1.98 | 0.15 | 2.72 | 0.19 | −3.02 | 84 | $p < 0.003$ | $r = 0.31$ |
| Thai PR * | 1.33 | 0.09 | 1.93 | 0.11 | −4.21 | 84 | $p < 0.001$ | $r = 0.42$ |
| PR total * | 2.58 | 0.12 | 3.33 | 0.13 | −4.24 | 84 | $p < 0.001$ | $r = 0.42$ |
| Melodic singing ability | 5.72 | 0.24 | 6.23 | 0.21 | −1.60 | 84 | $p = 0.11$ | $r = 0.17$ |
| Rhythmic singing ability | 6.57 | 0.18 | 6.96 | 0.18 | −1.53 | 84 | $p = 0.13$ | $r = 0.16$ |
| AMMA tonal | 24.95 | 0.72 | 26.73 | 0.81 | −1.63 | 84 | $p = 0.11$ | $r = 0.18$ |
| AMMA rhythm | 27.83 | 0.68 | 29.52 | 0.60 | −1.87 | 84 | $p = 0.07$ | $r = 0.20$ |
| STM | 14.45 | 0.58 | 15.98 | 0.58 | −1.87 | 84 | $p = 0.07$ | $r = 0.20$ |

* Remained significant after Benjamini–Hochberg correction for multiple testing ($p < 0.05$).

*3.4. Statistical Results 3: Interactions between the Musical Status and the High and Low Melodic Language Perceivers on the Language Performance Tasks (Two-Way ANOVA)*

A two-way ANOVA was performed to provide information about the role of musical status and melodic language perception with regard to the dependent variable PR total score. In addition, we wanted to provide information about the level of musical status represented in the low and high melodic LPs. In the group of the high melodic LPs, 18 participants were classified as musicians, 13 as amateurs, and 13 as non-musicians, while in the low melodic LPs, 12 classified themselves as musicians, 8 as amateurs, and as 22 non-musicians.

The results of the two-way ANOVA revealed a significant main effect for melodic perception, showing a group difference in the PR total score between the participants who perceive languages as more melodic/less melodic, respectively ($F(5, 80) = 13.17$, $p = 0.001$, partial $\eta^2 = 0.14$: high melodic LP ($M = 3.33$, $SD = 0.87$) and low melodic LP ($M = 2.58$, $SD = 0.75$)).

There was also a significant main effect for musical status ($F(5, 80) = 11.67$, $p = 0.001$, partial $\eta^2 = 0.23$). As we had unequal group sizes, Gabriel-corrected post-hoc analysis was applied. The results revealed that the professional musicians ($M = 3.55$, $SD = 0.81$) performed significantly better than the amateurs ($M = 2.84$, $SD = 0.70$) and the non-musicians ($M = 2.54$, $SD = 0.79$) in the language performances. This difference could not be observed between amateurs and non-musicians. However, there was no significant interaction between the musical status and melodic language perception ($F(5, 80) = 0.34$, $p = 0.72$; partial $\eta^2 = 0.008$). The effect sizes of the high and low melodic perceivers was $f = 0.40$, and the musical categories was $f = 0.55$. Both represent a large effect.

## 4. Discussion

In this research, we wanted to investigate the melodic perception of unfamiliar speech and to provide information about whether individuals who perceive languages as more melodic than others perform better than those who do not (Q1). Furthermore, we looked at whether the melodic perception of speech is another predictor that contributes to the ability to pronounce unfamiliar languages beside familiar ones (Q2). Finally, we also wanted to show whether musical abilities and musical status (professionals, amateurs, and non-musicians) are also connected with the melodic perception of speech (Q3).

*4.1. Correlational Analysis and Regression: Pronunciation*

We ran correlational and regression analyses to examine possible interdependencies between our variables and to address our first two research questions (Q1 and Q2). The main correlational analysis was based on the composite scores of the language performance (PR) and melodic perception (melodic P) tasks as this represents a more reliable and robust concept (Christiner 2020). We also provided the correlations and regression models of the five individual languages in the supplement for further illustration (see Supplement Tables S1–S6).

The results revealed that the PR score is related to the melodic perception of the languages, STM capacity, the number of foreign languages spoken, educational status, and to all musical measures. Based on the correlations, we performed regression models like in our previous research (Christiner 2020). We entered all the main variables that correlated with the PR total score. The results showed that 59 percent of the variance of pronunciation ability was explained by five predictors: the number of foreign languages, STM capacity, tonal musical aptitude, how melodic the languages appeared to the individuals, and how well the participants were able to sing familiar and unfamiliar melodies. In previous investigations, four of these predictor variables were already found to explain individual differences in the ability to pronounce new words (Christiner 2020; Christiner and Reiterer 2013), while the melodic perception of language is indeed a new dimension.

Previous research illustrated that the number of foreign languages spoken has an impact on how fast new language material is acquired. Language learners of second or



third languages, such as successive and late bilinguals as well as polyglots, benefit from knowing larger phonological structures. Therefore, it has generally been accepted that early and late bilingualism or multilingualism can improve the ability to acquire new phonological forms (van Hell and Mahn 1997; Kaushanskaya and Marian 2009; Papagno and Vallar 1995). This together with an improved STM has been found to explain why some individuals show better language capacities than others (van Hell and Mahn 1997). Indeed, STM capacity, as measured by digit spans, predicts individual differences in the language performance measures in this investigation as well. This finding was expected, as digit spans measure the phonological loop capacity, which is seen as the criterion that is associated with foreign language success (Baddeley 2010; Biedroń and Pawlak 2016). Evidence of a positive relationship between singing ability and the learning, retrieval, and pronunciation of new languages has been provided in several investigations. It was shown that new word learning in foreign languages was most successful in a singing condition (Ludke et al. 2014). Additionally, singers were better at the pronunciation of new words than non-musicians (Christiner and Reiterer 2015). On the one hand, the reason why singing supports language acquisition processes seem to depend on the combination of language and melody, which facilitates memorization (Fonseca-Mora 2000; Gordon et al. 2010; Thiessen and Saffran 2009). On the other hand, singing ability may also help to retain new words since it has been suggested that the oromotor system assists memorization of speech sounds (Schulze and Koelsch 2012). This could explain why professional singers, who train their vocal motor system, outperform non-musicians and pure instrumentalists (Christiner and Reiterer 2015).

A positive relationship between pitch discrimination ability and language skills has been shown in several investigations (Christiner 2020; Christiner et al. 2018; Coumel et al. 2019; Moreno 2009). Thus, musical capacity facilitates the perception of pitch contour in spoken language (Schön et al. 2004) and enhances pitch processing in language (Marques et al. 2007). According to Patel's OPERA hypothesis (2011), five conditions lead to higher neural plasticity in speech-processing networks. Two of them are most crucial to the interpretation of our findings. These are overlaps between speech and music processing as well as precision. The first condition assumes that if musical training improves speech processing, there must be overlapping brain networks. Pitch sounds are mostly processed in the right-hemisphere, whereas left hemispheric processing of lexical tones is more likely induced by syntactic or semantic information (Patel 2011). Indeed, right-hemispheric specialisation has also been noted in the case of linguistic information that is rather poor in content (Perkins et al. 1996), which is one reason why musical abilities may predict the ability to pronounce unfamiliar words in this investigation. The language material was unintelligible to the participants and was processed in a similar way to pitch sounds. In the same vein, it can also be suggested that musicians and individuals who possess musical aptitude are more precise in encoding acoustic information in speech. Indeed, the second condition of the OPERA hypothesis, precision, suggests that music perception places higher demands on the encoding of specific acoustic elements (Patel 2011). This fine-grained musical ability may help in imitating unfamiliar languages more precisely since individuals who possess high musical aptitude may rely more on the acoustic components when trying to remember and imitate unfamiliar speech (Christiner and Reiterer 2015).

Interestingly, the musical measurements of AMMA consist of a rhythmical and a tonal part. In our regression model, the rhythmical criteria did not explain the variability of the language performances, neither in the composite condition based on the PR score nor for the five individual languages (see Table S6 in the Supplement). Indeed, this observation has already been made several times (Christiner 2020) (even though the opposite was also found (Swaminathan et al. 2017; Christiner et al. 2018). Learners of new languages have to be sensitive to the temporal and the tonal features of the target language. Individuals often fail to understand where words begin or end in sequences of spoken language (Patel 2007). Therefore, language learners need to be able to adapt to the intonational, rhythmic, and melodic aspects of the target language as accurately as possible. This may be one

reason why relationships between language performance and rhythmic and tonal musical aptitude are commonly found. The reason why tonal aptitude may predict language capacity more often than the rhythmic parameter may be that tonal aptitude has been said to predict productive phonology in general (Slevc and Miyake 2006). Some researchers have also suggested that the positive transfer from music to language is based on cognitive mechanisms of pitch processing, which are probably shared between music and language (Perrachione et al. 2013). Indeed, studies have revealed that individuals who possess more elaborate tonal aptitude are also better at detecting tonal variations in Mandarin (Delogu et al. 2006) and are also better at imitating short sequences of tone languages (Christiner et al. 2018; Christiner 2020). However, enhanced pitch perception ability has also been linked to the pronunciation of a number of non-tone languages such as English, Spanish, Farsi, Tagalog, and Japanese (Milovanov et al. 2009; Posedel et al. 2012; Christiner 2020). This illustrates that tonal capacity is related to language ability regardless of language typology.

*4.2. Melodic Perception of Languages and Performance*

The melodic perception of languages is a new predictor when it comes to the ability to retrieve and pronounce new words. Participants were instructed to indicate their subjective impression of how melodic and musical the spoken languages sounded to them. We found correlations between language performances and melodic perception. In addition, the melodic perception of the languages turned out to predict the variance in the pronunciation skills in the main regression model. This finding was similar to that of a cluster analysis based on the melodic perception of the five languages where we could detect two distinct groups (see Table S7). One group perceived all five languages as more melodic than the other. The high melodic LP was also the group that performed significantly better in the language performance tests.

The impact of melody on language learning capacities has been illustrated in various ways. Melodic perception certainly plays a crucial role from early childhood on in music and language acquisition processes when infants learn their first language(s) and any kind of music. Vocalizations by parents targeted at infants are often rather slow, show more pitch variations, and are more melodic in their characteristics than normal speech, which makes it easier for infants to learn their mother tongue (Kuhl et al. 1997; McMullen and Saffran 2004). Melody has also been said to facilitate the memorization and retrieval of words (Rainey and Larsen 2002), and new utterances are more easily learnt if they are sung (Gordon et al. 2010; Ludke et al. 2014; Wallace 1994). The reason for the positive relationship between memory and singing has been related to the impact of melody on cognitive functions (Purnell-Webb and Speelman 2008), since melody probably serves as a mnemonic with which utterances are stored (Rainey and Larsen 2002). In this investigation, we were able to show that the subjective impression of the melodic perception of different languages improves taking up new languages as well.

*4.3. Musical Abilities, Musical Status, and the Melodic Perception of Speech*

To uncover any potential relationship between melodic perception of languages and musical measures (singing, musical aptitude, and musical status) we ran a number of statistical tests. We performed *t*-tests to find out whether the musical dimensions of the AMMA test and performance in the singing tasks were significantly different between the high and low melodic perceivers (Table 5). In addition, we performed an ANOVA, in which musical status was our grouping variable and melodic perception of the languages our dependent variable (see Supplement Tables S8 and S9). Then, since we had found that language performances were significantly better among high melodic perceivers and musicians, we also performed a two-way ANOVA (Section 3.4) to uncover possible interactions between musical status (professionals, amateurs, and non-musicians) and melodic perception (high and low melodic LPs). The results indicated that the relationship

between the musical measures of this investigation and the melodic perception of languages was inconsistent.

The melodic perception of languages, as investigated in this research, contributes to the ability to pronounce new languages. We provided only little evidence that musical capacity is related to how melodic languages appear to individuals. Research has illustrated that the repetition of speech can transform language to sound like song (Deutsch et al. 2011). Margulis et al. (2015) used this speech-to-song illusion and found that languages that were more difficult to pronounce appeared to be more musical even before any repetition took place. This shows that individual differences in how musical languages appear may be influenced by various factors that are not necessarily related to acoustic elements.

## 5. Conclusions

The results of this investigation show that how accurately new languages are pronounced depends on several cognitive skills. We found that the more languages individuals spoke, the more accurately they pronounced the unfamiliar languages. The same was true for STM capacity, which was also enhanced in individuals who possess elaborate pronunciation skills. In addition, our findings indicate that musical ability predicts individual differences in taking up new languages. Tonal aptitude and the ability to sing melodies predicted well individual differences in pronunciation skills. The findings of this study also add a new dimension to research on individual differences by showing that individuals who perceive languages as more melodic than others also retrieve and pronounce utterances more accurately. We speculated that musical abilities could be responsible for the extent of melodic language perception but found only little evidence. Except for a few correlations between musical aptitude and the melodic perception of languages, none of our other musical measures offered any link to how melodic the languages sounded to our participants. Future directions may include an acoustic analysis of why particular natural languages are perceived to be more melodic and tuneful than others. Since speech can also be turned into song by repetition of utterances, factors outside the acoustic domain and its relationship to the melodic perception should be investigated as well. In this respect, sociocultural and sociolinguistic approaches should also be included to reveal what shapes an individual's capacity to perceive languages in melodic terms.

**Supplementary Materials:** The following are available online at https://www.mdpi.com/article/10.3390/languages6030132/s1, Figure S1: Lyrics and notes of the new song learning task, Tables S1–S5: Correlations of the individual languages, Table S6: Regression models of the individual languages, Table S7: Cluster analysis and melodic ratings, Table S8: Welch's F-test ANOVA musicians, Table S9: Games–Howell post-hoc analyses.

**Author Contributions:** Conceptualization, M.C. and C.G.; Data curation, M.C., C.G. and A.S.-P.; Formal analysis, M.C. and C.G.; Investigation, M.C. and C.G.; Methodology, M.C. and C.G.; Project administration, M.C. and C.G.; Resources, M.C.; Supervision, P.S.; Writing—review & editing, A.S.-P., M.C., and C.G. All authors have read and agreed to the published version of the manuscript.

**Funding:** M.C. is funded within the Post-DocTrack Programme of the OeAW. Open Access Funding by the University of Graz.

**Institutional Review Board Statement:** The study was conducted according to the guidelines of the Declaration of Helsinki, and approved by the Institutional Review Board of the Medical Faculty of Heidelberg S-778/2018.

**Informed Consent Statement:** Informed consent was obtained from all subjects involved in the study.

**Data Availability Statement:** Data is contained within the article or supplementary material.

**Acknowledgments:** The authors acknowledge the financial support of the University of Graz. Our gratitude also goes to Angelika Rumpler for her valuable comments on language.

**Conflicts of Interest:** The authors declare no conflict of interest.

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
