# Peer review of "The Melody of Speech: What the Melodic Perception of Speech Reveals about Language Performance and Musical Abilities"

_languages, doi:10.3390/languages6030132_

Round 1

Reviewer 1 Report

The degree to which melodic processing of an unfamiliar language is related to language production is a theoretically interesting question and is addressed in the current manuscript. The authors also discuss how individual differences in a number of different factors relate to foreign language production. Although the research question is theoretically interesting, overall, I found the organization of the manuscript difficult to follow. There were multiple instances in the Introduction and Discussion that needed to be further developed and framed more clearly with respect to previous research. I also found the organization of the methods section to be confusing. It could be helpful to include a table with demographic information for participants. For the participants section, if not including a table, I recommend discussing music and language experience in this section. I also found the results section to be very long and the justification for running certain analyses was not clearly provided, particularly for results that were redundant with earlier analyses. I provide a few specific comments below.

Introduction - First paragraph of Intro is framed rather vaguely. Also make sure to check APA formatting for citations throughout the Introduction and Discussion.

Third full paragraph of Intro – Need to provide more context for WM and STM discussion. The authors state, “In research on memory of music and language a further cognitive ability, working memory (WM) capacity, has received considerable attention.”, but the authors omit a discussion of music and working memory and only focus on language.

Pg 3 line 137- In general, I would recommend changing “unintelligible” to “unfamiliar”

Participants – minor – avoid starting sentence with number and when needed, use word to express a number if it begins a sentence.

Participants – minor – Change “age mean” to “mean age” and include standard deviation.   

Participants – minor – I recommend restructuring the following sentence,  “In addition, the participants reported the number of instruments they played, while 35 participants played no musical instrument, 22 played one, 18 played two, 2 played three, 4 played four, 5 played five, and 1 played seven.”

Second singing task – For this task, it wasn’t clear to me why lyrics would be included as this adds a potential confound of verbal STM demands.

2.4.2 I would like more information on how language raters were selected

Table 1 – minor – italicize p and effect size

Pg 9 line 398- minor – remove first comma in sentence

Table 2 – minor – need effect size for all comparisons. Change commas to decimal points.

Figure 1 is blurry and as a result is difficult to read.

Pg 12 line 480 – missing effect size

Results – In general, I found the results section to feel disconnected and is also very long. I would like to see this section re-organized so that the effects of primary interest are given more focus than effects of lesser theoretical interest.  Additionally, there is redundancy across the analyses, adding to unnecessary length of the section. I would also recommend moving the descriptive statistics to a table.

Results –Chi-square - I’m not clear on what is being measured by the factor gender. Is this speaker gender or participant gender? I would like more justification for why this analysis was run.

Results – regression models – need explanation for ordering of variables. Are the results dependent on that ordering or is the same pattern found for different ordering?

Results –Many of the variable names are confusing. I would recommend renaming the variables to more clearly reflect the measured construct and/or stating what measure the variable corresponds to in a note under the table that mentions the variable.

Pg 14- line 539 – I found the phrase “melodic sensation” to be confusing, I would prefer “melodic perception” or would like this phrase clarified.

Discussion -  The link between melodic perception and language performance can be presented more clearly. In general, throughout the discussion I would like the interpretation of the results to be more developed.   

Author Response

 First of all, we want to say thank you for your valuable comments and input we received. We hope to have addressed your questions satisfactorily. We wrote our responses question by question and included the line numbers. We hope this reduces your workload.

Reviewer 2 Report

Summary: This paper would be of interest to a broad audience of music science and language science researchers and would especially fit well within the speech-to-song illusion literature, even if the authors do not study the speech to song illusion directly. The speech-to-song illusion literature is interested in what acoustic/semantic features lead spoken utterances to sound musical (not just melodic; a term change to musical is one that the authors might think about adopting). As the paper stands now, I did not understand the motivation for the questions the authors addressed in the paper, several key pieces of the literature were not addressed, and the statistical analyses did not fully tell the story of their data. I suggest a complete reframe of the paper to more properly motivate the study that they did do and different analyses to directly characterize the contribution of each language to production outcomes in addition to the analyses in the paper characterizing what factors predict melodic rating/foreign language pronunciation. A lot of the relevant literature is cited in the paper, but it is not organized in a way that properly motivates the questions that their data can address. I applaud the work that the researchers did in carefully thinking through the questions that their work might address, the data has the potential to address significant gaps in the literature related to the perception of musicality, musical training, and language processing. With some reframing and new analyses this work will make a significant contribution to the field.

Abstract: It’s not clear what is meant by a musical ear toward the end of the abstract. Is this a proxy for musical training? Or Musical aptitude as measured by the MET (Musical Ear Test)? It seems contradictory to all the positive relationships with tonal aptitude and melodic singing ability from above. That is, in one sentence you highlight all the ways that individual differences matter and in the next sentence say that perceiving musical features in speech is just a general human trait (not subject to individual variability…).

Introduction

Lines 31-35: I think that Lehrdahl and Jackendoff, 2006 (which you cite below!) or Jackendoff, 2009 would be better citations for these big foundational connections between music and language instead of Limb, 2006 (or at least alongside Limb, 2006). It would also be helpful to use more specific terms instead of “properties” or “principles”. As I read the first paragraph I find myself asking “which properties are shared between music and language? Or “what are the shared fundamental principles that lead to music and language both adopting hierarchical structures?”

Lines 45-51: since you don’t find effects of musical ear it would be a good idea to clearly define aptitude vs. musical training (see Swaminathan & Schellenberg, 2020 or Kragness et al., 2020) and to discuss the other side of the story, that is, that musical training/aptitude sometimes do not predict language abilities.

Lines 57-59: citation needed

Lines 73-75: Please cite/summarize similar studies with different outcomes such as Swaminathan and Corrigal

Lines 79-86: I am not sure I understand the link from language and music comparisons and relationships between linguistic and music abilities and the jump to the main question, which is about how an individual’s perception of melodic (or musical?) features of speech could be related to language learning/perception/production outcomes. A survey of the skills that are important in language learning (e.g., word segmentation in foreign languages can be made easier by attention to prosodic stress or other musical features of speech, fine grained pitch processing in music could be important for novel linguistic sound processing – vowels in particular, etc). As it stands now, there is one uncited reference describing Spanish as very musical and this doesn’t motivate the question the authors are appearing to address. It also sets up the reader to anticipate that certain languages over others are perceived as more or less musical, instead of asking whether individual differences in the perception of musicality in a single language will predict language performance. I now see that the authors have used multiple languages for their outcome measure, which really changes the narrative/rationale of this study for me. That is, the question that should be addressed should be whether or not how melodic a given language is perceived by an individual is predictive of their pronunciation in that language.

Results

Participants “melodic” ratings of speech should be submitted to a repeated measures ANOVA with each language as a level of the main variable Language. Then we can understand whether or not certain languages over others are perceived as more or less musical. There is some work on this question already in the literature from Margulis et al., 2015 “Pronunciation difficulty, temporal regularity, and the speech-to-song illusion” and it would be helpful to reference this and other speech-to-song illusion work in the introduction to better motivate the research you actually did perform (see Vanden Bosch der Nederlanden et al., 2015; Tierney, Patel, & Breen, 2018).

Cluster analyses – please plot a scatterplot of all data with points from each of the two groups represented in different colors so that the reader can assess how the cluster analysis created the groups (e.g., was it a truly bimodal distribution? was there a lot of overlap? would a median split be equivalent to the solution the cluster analyses came to?) There is also a whole section comparing professionals/amateurs/non-musicians. I would suggest choosing one clustering approach and sticking with that. The other analysis can be included in the supplement. There is too much data to digest easily in the results section.

Tables should mostly be replaced by figures, and in text references of mean and standard deviation could be streamlined or only referenced in a succinct table. See Swaminathan & Schellenberg, 2020 or Vanden Bosch der Nederlanden et al., 2018 for examples of how to display lots of different data used for regressions in a succinct format.

Figure 1 should be a stacked bar or perhaps part of the participants section in the method as a table.

Many of the results could be summarized more succinctly. For instance, 3.4 statistical results 3 could be a single sentence “There were no differences in participants musical training background between the two groups, chi-square results, or in terms of participants sex, chi-square results.” Please also report full p-values instead of p > .05 unless the p value is p>.001. For example, Table 4 should have all significance values reported with exact numbers. Also, the upper and lower confidence interval is not typically reported, instead the effect size (d or eta-squared).

Abbreviations are used throughout the document that are hard to follow. I would suggest either not using these abbreviations, or having a section in the method where every single abbreviation is listed. Each table or figure caption should also repeat the abbreviations so that it is easier for the reader to understand what each variable is in the results section.

Please display the correlation matrix. Is musical training not correlated with any of the variables? It would be important to put these variables in the model even if they don’t correlate with the outcome measure in order to be able to say that musical training is not a significant predictor. Including it as a variable will also allow you to control for that variable while understanding whether the other variables still remain significant predictors in the model even after accounting for musical training, etc.

There are many non-parametric tests included in the analysis section that seem like they should be parametric. Please either use parametric tests or justify why the non-parametric are necessary.

Since there are many issues with the analyses, the discussion does not logically follow from the reported results and so I do not provide specific comments on this section. However, there is a lot of new literature cited here that should probably be better summarized in the introduction to prepare the reader for what possible outcomes they could expect from your experiment.

Author Response

(The authors gave the same response as above.)

Reviewer 3 Report

A brief summary (one short paragraph) outlining the aim of the paper and its main contributions.

The authors conducted a study to examine the relationship between participants’ perceptions of melody within unfamiliar languages and their pronunciation of these languages, musical abilities, and short-term memory. Two analytical approaches (group comparisons based on cluster analysis and multiple regression) support the primary result that suggests that language melody perception influences language pronunciation, and that this relationship was at least partly independent of musical expertise.

Broad comments highlighting areas of strength and weakness. These comments should be specific enough for authors to be able to respond.

Strengths

The premises of the research are intriguing and warranted. I think many language and music cognition scholars will be interested in this paper, especially because it suggests important new directions for investigation.

The research questions are clearly articulated and reasonably well-motivated. I appreciate how the authors clearly organized their paper based on the research questions.

The writing quality is good overall, with only moderate organization issues and minor punctuation and spelling errors (see specific comments).

Weaknesses

The authors make little effort to define what they mean by speech/language “melody” in the Intro, nor do they effectively operationalize it with the Method. Many of the important results that compare low and high language melody perceivers are thus based on tenuous ground. At the end of the Discussion, the authors seem to admit that they (or their participants) are not really sure what speech/language melody represents. I would expect participants’ melody perception ratings to correlate with particular acoustic aspects of the language stimuli, but no attempt is made to perform such a task. Furthermore, the paper lacks any discussion of tonal languages and their relationship to speech melody—this seems like a substantial omission.

Some procedural details are confusing or incomplete, which would make it difficult to systematically replicate this study. For example, the coverage of the song and language production ratings leaves many questions unanswered (see specific comments).

Specific comments referring to line numbers, tables or figures. Reviewers need not comment on formatting issues that do not obscure the meaning of the paper, as these will be addressed by editors.

  1. 2, lines 52-59: Because short-term memory constitutes a major aspect of the paper, the paragraph on memory could use a few more sentences (with citations) to situate working memory and short-term memory within the broader memory system.
  2. 2, lines 79-86: Given that the authors are studying language, not music, they must define what they mean by “melody.” In this paragraph, the authors refer to an idea from Grillparzer (“Spanish is the most melodic language”). However, there is not enough context to evaluate this claim. For example, within the musical domain, melody and rhythm are widely viewed as distinct. This may not be true within language because, due to prosodic variation, a listener may perceive a sentence as melodic, even when it is monotone.
  3. 3, lines 125-126: The authors point to a potential weakness (relative pitch errors) of computerized methods of singing analysis. However, this is not a genuine weakness of the computerized approach because it is actually possible to account for relative pitch errors in computerized methods. Furthermore, it is not clear how/why subjective measures are immune to the stated weakness. Thus, rather than point to the potential weaknesses of the computerize approach, it may be better to articulate the strengths of the subjective approach.
  4. 4, lines 165-180: Please clarify your predictions, if applicable. Note: some predictions appear later in section 2.7, but they would probably be better placed at the end of the Intro instead of within the Method.
  5. 4, lines 171-172: The authors say that “language typology … will not be investigated in detail” but they clearly do compare performance across languages in the Results. Please clarify your meaning here.
  6. 4, line 186: Please provide SD alongside M. This applies throughout the paper (e.g., see p. 12, section 3.4).
  7. 5, lines 209-215: Please indicate which AMMA subtests were used.
  8. 5, lines 221-230: Please provide more information about the song imitation stimuli (e.g., share/show the stimuli). Consider also that the task was likely harder for some participants simply because of the tonal range. Participants who have to transpose to their own register may perform worse due to the cognitive resources required for the transformation. Transpositions may have affected the subjective ratings, too.
  9. 5-6, lines 231-246: I have many questions about the singing ability ratings. Were the raters blinded to the purpose of the study, the aptitude of the participants, etc.? Did the raters evaluate the song performances independently (e.g., in isolation)? Did the raters rate all of the productions, or did they split them? What proportion of songs were utilized to determine reliability? Were raters’ scores averaged in any way? Why did you combine the familiar and imitation ratings into compound measures (were the ratings correlated)? Why did you use Cronbach’s alpha, a measure typically associated with internal reliability, to measure interrater reliability (instead of a series of rater-pairwise correlations)?
  10. 6, lines 264-266: Please explain how you ensured that speech acoustic features were constant across the language production stimuli. Please clarify the acoustic variation that was left open for language/speech melody (e.g., intonation?).
  11. 7, lines 284-286: It’s not clear how you calculated Cronbach’s alpha for interrater reliability. Please clarify (I had a similar question about this statistical application for the song production ratings).
  12. 7, lines 288-296: This paragraph suggests that the participants were not informed about what “melodic” means for the purpose of the rating. If that is the case, then it complicates the authors’ abilities to make meaningful inferences with these data—especially because the authors later use the data to categorize participants into low- and high-melodic language perceivers. Secondly, please clarify whether each of the four samples (on each trial) were the same or different. If they were the same, then that would probably be enough to activate the speech-to-song illusion (see work by D. Deutsch and others), which would surely influence the results (consider that some languages/sentences/phrases may be more likely to induce the illusion and thus lead to higher melodic ratings).
  13. 7, lines 298-302: Please clarify the parameters of your STM measure. What were the min/max digits? Was stimulus presentation automated? Were digit stimuli randomized across participants?
  14. 7, line 310: was this a laboratory study or an online study? It’s not clear how participants were directed to the “online platform” nor is it clear how much interaction/instruction they had with the experimenters (e.g., to explain each task and answer questions). I recommend that the authors write more content within this section (2.6) to explain the research environment.
  15. 8, line 337: Please clarify whether the “pronunciation total score” was simply the sum of raters’ scores for all of each participant’s productions. If so, the simple sum should only be used if every rater rated every production. Sums should be avoided if there was uneven distribution of productions across raters.
  16. 8, lines 345-346: Min/max values are provided for Russian but not for the other languages (I don’t think min/max scores are very useful, but if you are going to include them, please be consistent).
  17. 8, lines 364-365: What does “the mean of the number of foreign languages” represent? If this is just a demographic item, it need only be covered in the Participants/Language background section. It becomes clear later (e.g., p. 9) that the authors are reporting descriptive statistics for other background variables (e.g., number on instruments played, educational background—which should not be averaged because it is a categorical variable). These background demographic data seem out of place in the Results, especially because the values are also provided in the Materials and Methods section. I recommend you move all background variables to the Materials and Methods section.
  18. 9, line 376: Please provide appropriate units for the result (e.g., here the authors presumably mean “15.23 digits”).
  19. 9, lines 385-388: I think it is possible that the difference between low and high groups is explained by non-melodic perception factors, but this is not addressed. For example, some participants may be biased toward higher ratings, regardless of speech melody perception. Without some practice, training, or otherwise clear instructions for the melodic rating task, it’s difficult to interpret these group categories. I agree with the authors that the categories reflect a “general phenomenon” (p. 10, line 414) but I’m not sure it is explained by melodic perception.
  20. 10, Table 2: Please report effect sizes and p values for all comparisons (regardless of statistical significance). Please be consistent with decimal symbol (. or ,). Note that Table 5 only uses commas.
  21. 10, Table 3: Please report effect sizes and p values for all comparisons (regardless of statistical significance).
  22. 11, lines 439: The authors refer to ANOVAs in section 3.3.3, but that section doesn’t appear to exist. The ANOVAs they refer to cannot be those from Table 2 because those included three groups (not two). I presume the authors mean section 3.4.X instead.
  23. 11, lines 454-460: Please reword your null results because a nonsignificant result does not provide evidence for the nil hypothesis (see Cohen, 1994, The Earth is Round). Therefore, you cannot claim “no association” but instead something like “no support for an association.” Next, please provide effect size for each chi-squared test for independence (e.g., Cramér's V). Also, please provide exact p values throughout the paper (except when they are very small, e.g., < .001).
  24. 12, Figure 1: Why don’t all of the percentages sum to 100?
  25. 12, line 474: p values can never equal 0. Presumably the authors meant p < .001.
  26. 12, line 480: Please provide effect size.
  27. 14, lines 528-529: Do the authors mean “typologically different” or “typologically similar” or something else?
  28. 14, line 553: I don’t think this study provided any evidence for the claim that high melodic LP persons “memorize” new utterances better.
  29. 16, lines 623-639: The discussion could be bolstered with more coverage of relevant relationship between musical and language abilities with special emphasis on speech processing (e.g., Patel’s OPERA hypothesis).
  30. 16, line 649-650: What do the authors mean by writing that melodic perception of languages “seems to be a rather individual concept?” I read this as a weak admission about not knowing what their language melody perception task actually measures. I agree that this is a problem, so more effort should be taken to develop this point in the Discussion.

General comment: There are minor punctuation (e.g., missing commas) and spelling (e.g., social identify --> social identity) errors in the paper that should be corrected.

Author Response

(The authors gave the same response as above.)

Round 2

Reviewer 1 Report

I appreciate the authors’ revisions to the manuscript. Overall, I am satisfied with the improvements that have been made regarding the organization and overall clarify of the paper. I have a few remaining comments for the authors to consider. In general, I recommend removing the word “prove” or “proven” from the manuscript. I would also caution the authors to avoid using causal language when discussing results based on correlational analyses. I outline my remaining concerns below.  

Minor – pg. 1 line 40 -Need citation for Patel (2008) in Reference section

Minor – pg 2 line 72 – the phrase “WM has been found to enhance different cognitive abilities” is now redundant with the beginning of the paragraph. I suggest deleting the phrase and changing the previous sentence to “Therefore, WM capacity has received considerable attention in music and language research and is associated with individual differences in the mastery of first and foreign languages (Baddeley, Gathercole and Papagno 1998; Dörnyei and Ryan 2015; Majerus et al. 2006; Wen and Skehan 2011).

Minor – pg 3 line 119 – change “serious” to “series”

Minor – pg 6 lines 266-267 – change “und” to “and”

Minor – pg 7 line 323 – I wasn’t clear on how to interpret the phrase “(No of FL)” until a couple pages later when I read Table 1. To reduce the confusion for the reader, I would recommend revising the beginning of the paragraph to include phrasing like, “Given participants’ experience with foreign languages, we addressed the number of foreign languages (No of FL) spoken by participants in the study.  Results have shown that 10 participants did not speak a second language, even though they were taught English in school, 28 participants reported to speak one foreign language, 36 indicated to speak two foreign languages, and 12 participants reported to speak three foreign languages”

Minor – pg9 line 409 – recommend changing “one of our supervisors” to “an experimenter”

Minor – Table 1 – I don’t believe “No of MI” is used in the paper other than in the table. For this reason, “No of MI” should be deleted from Table 1.  

Results – the variable names are not being consistently used. For instance, in Table 3 the authors state “AMMA tonal” rather than “AMMA T” and “Melodic singing ability” rather than “Melodic SA”. I prefer the less ambiguous former terms rather than the latter ambiguous abbreviations. However, either way, the variable names should be consistently used throughout the manuscript. 

Minor – italicize statistics throughout paper

The supplemental figures are blurry and cannot be read. 

Results – please include how many low and high melodic language perceivers were represented at each level of musical status. 

Pg 15 – line 582 – It’s not clear to me what the authors mean by “rhythm seems to have an effect on language performance” when interpreting a bivariate correlation. As written, the sentence reads as a causal claim, which cannot be made when relying on correlational analyses. 

Discussion – While the authors detail their results pertaining to the measure of rhythmic musical aptitude, I believe the discussion section would benefit from a more detailed discussion of the results relating to their measure of tonal musical aptitude, particularly with respect to how these results connect to the previous paragraph where the authors discuss the OPERA hypothesis.  

Pg 15 line 589 – avoid making causal claims based on regression analysis. 

Pg 16 line 621 – I found the following sentence to be a bit misleading as written, “Results indicated that none of our approaches illustrated a relationship between the musical measures and the melodic perception of languages, except for a few correlations (see tables S1, S2 and S4)” I believe the statement can be read as overgeneralizing the results. Even if the authors were to state that the relationships between musical measures and melodic perception of language are inconsistent, I believe that would be more appropriate.

Author Response

Dear reviewer, we want to thank you for your valuable comment and also for your precise instructions. Thanks to your feedback the manuscript improved remarkable.

Yours sincerely,

the authors
